# The Influence of Calcium on the Growth, Morphology and Gene Regulation in *Gemmatimonas phototrophica*

**DOI:** 10.3390/microorganisms11010027

**Published:** 2022-12-22

**Authors:** Sahana Shivaramu, Jürgen Tomasch, Karel Kopejtka, Mohit Kumar Saini, Syed Nadeem Hussain Bokhari, Hendrik Küpper, Michal Koblížek

**Affiliations:** 1Laboratory of Anoxygenic Phototrophs, Institute of Microbiology of the Czech Academy of Sciences, Novohradská 237, 37981 Třeboň, Czech Republic; 2Department of Plant Biophysics and Biochemistry, Institute of Plant Molecular Biology, Biology Centre of the Czech Academy of Sciences, Branišovská 1760/31, 37005 České Budějovice, Czech Republic; 3Department of Experimental Plant Biology, Faculty of Science, University of South Bohemia, Branišovská 1760/31a, 37005 České Budějovice, Czech Republic

**Keywords:** *Gemmatimonas phototrophica*, anoxygenic phototrophic bacteria, horizontal gene transfer, calcium, transcriptomics

## Abstract

The bacterium *Gemmatimonas phototrophica* AP64 isolated from a freshwater lake in the western Gobi Desert represents the first phototrophic member of the bacterial phylum Gemmatimonadota. This strain was originally cultured on agar plates because it did not grow in liquid medium. In contrast, the closely related species *G. groenlandica* TET16 grows both on solid and in liquid media. Here, we show that the growth of *G. phototrophica* in liquid medium can be induced by supplementing the medium with 20 mg CaCl_2_ L^−1^. When grown at a lower concentration of calcium (2 mg CaCl_2_ L^−1^) in the liquid medium, the growth was significantly delayed, cells were elongated and lacked flagella. The elevated requirement for calcium is relatively specific as it can be partially substituted by strontium, but not by magnesium. The transcriptome analysis documented that several groups of genes involved in flagella biosynthesis and transport of transition metals were co-activated after amendment of 20 mg CaCl_2_ L^−1^ to the medium. The presented results document that *G. phototrophica* requires a higher concentration of calcium for its metabolism and growth compared to other *Gemmatimonas* species.

## 1. Introduction

The bacterial phylum Gemmatimonadota was established by Kamagata in 2003 [1]. Its members are widespread and have been identified in diverse environments, such as soils, fresh waters, and sediments [2,3]. Until recently (2014), it contained only a handful of cultured heterotrophic species such as *Gemmatimonas aurantiaca* [4], *Gemmatirosa kalamazoonensis* [5], *Longimicrobium terrare* [6], and *Roseisolibacter agri* [7].

The first phototrophic member of the phylum, *Gemmatimonas phototrophica*, was isolated from a freshwater lake Tiān ér hú in the western Gobi Desert in Northern China [8]. It is a facultative photoheterotroph that requires oxygen for its growth and bacteriochlorophyll (BChl) synthesis [9,10]. *G. phototrophica* contains unique photosynthetic complexes with a double concentric ring antenna surrounding type-2 reaction centers (RC) [11,12]. Its genome contains a complete set of genes encoding RC and genes for BChl *a* biosynthesis organized in the photosynthesis gene cluster (PGC). The genes found inside the PGC of *G. phototrophica* are closely related to the genes from Proteobacteria, which suggests that Gemmatimonadota receive their PGC via horizontal gene transfer (HGT). Therefore, Gemmatimonadota is the only known phylum with a documented acquisition of the entire PGC via HGT event from a distant bacterial phylum [8,13].

Until recently, we cultured *G. phototrophica* on agar plates under semiaerobic conditions. Under these conditions it grew slowly with generation times between 2 and 4 days [10]. All the attempts to establish liquid cultures in the past few years have failed. This was quite puzzling, considering the fact that members of the *Gemmatimonas* genus seem to be mostly planktic species common in temperate freshwater lakes [14,15]. Moreover, the closely related species *G. aurantiaca* and *G. groenlandica*, as well as all other cultured Gemmatimonadota species, grow in liquid medium [15,16].

Growth is one of the main attributes that provides an ultimate measure of the metabolic activity of all living organisms [17]. Although bacteria have considerable metabolic versatility, they typically require only a limited number of macronutrients for growth, providing major elements (carbon, hydrogen, nitrogen, oxygen, phosphorus, and sulfur) as well as trace minerals, vitamins, or other growth factors. The challenge in this case was to establish how to provide the right substrates at optimal concentrations under the best physico-chemical conditions to allow the growth in liquid medium. The lack of established liquid culture under laboratory conditions significantly restricted further research on *G. phototrophica*. Basic experiments documenting phototrophic capacity were conducted with cells grown on agar plates (solid medium) [8,10], but many experiments became problematic as the agar-grown cultures suffered from poorly defined physiology and growth characteristics. Therefore, we decided to find suitable growth conditions for *G. phototrophica* in liquid medium.

## 2. Materials and Methods

### 2.1. Strains, Media and Growth Conditions

The experiments were conducted with the bacterium *G. phototrophica* AP64 (DSM 29774^T^). The previously described growth medium R2A^+^ was used to culture the cells on agar plates [8]. However, this medium was modified in order to test various nutrients to increase the yield of biomass. Different agar sources like Bacto^TM^ Agar (BD Biosciences, Sparks, MD, USA), Plant agar (Duchefa Biochemie, Haarlem, The Netherlands) and Micro agar (Duchefa Biochemie, Haarlem, The Netherlands) were assayed. Different carbon sources, vitamins and microelements were tested for the growth of cells on agar plates. Following the results obtained on agar plates, the growth experiments were performed with liquid media. Initially, the cells were streaked on agar plates and incubated for two weeks in microaerophilic conditions (10% O_2_ + 90% N_2_) at 28 ± 1 °C, and pH 7.7 in the dark. The individual colonies were picked from the agar plates and then transferred to liquid medium for continuous cultivation. The cells were grown in liquid medium containing (L^−1^) 0.5 g yeast extract, 0.5 g peptone, 0.3 g pyruvate, 0.5 g glucose, 0.5 g soluble starch, 0.3 g K_2_HPO_4_, 1 mL of modified SL8 trace metal solution (mL^−1^: 190 µg CoCl_2_.6H_2_O, 5.2 mg Na_2_-EDTA, 24 µg NiCl_2_.6H_2_O, 17 µg CuCl_2_.2H_2_O, 70 µg ZnCl_2_, 20.3 mg MgCl_2_, 62 µg H_3_BO_3_), and 1 mL of vitamin solution (mL^−1^: 200 µg B1, 20 µg B3, 10 µg B7, 10 µg B12) at 28 ± 1 °C and pH 7.3. If not stated otherwise, the concentration of CaCl_2_ used to obtain liquid cultures was 20 mg L^−1^. Cultures were incubated aerobically in Erlenmeyer flasks with cotton plugs on an orbital shaker (150 RPM) at 25 ± 1 °C in the dark for 8–10 days. At the beginning of each experiment, 1 mL of the inoculum (approx. OD_600_ = 0.25) was diluted in 100 mL of a fresh medium (equivalent to 1% *v/v* to the fresh nutrient media). The nutrient tests were made in duplicates depending upon the nutrient being tested. The effect of different nutrient sources on the growth were tested for both the presence and absence of the corresponding substrate. The growth response of *G. phototrophica* was tested and confirmed with semi-liquid agar slants and full liquid media at different concentrations of oxygen (atmospheric: 100%, 80%, and ∼0%, *v*/*v*) in the headspace of the medium. The liquid media were supplied with the corresponding mixture of sterile nitrogen and oxygen. In order to determine the optimal growth temperature, the cells were grown in the liquid media over the temperature range of 25–30 °C (± 1 °C). The optimal pH was tested in liquid media with pH ranging from 7.3 to 8.0 at 25 °C (± 1 °C). The pH values of the medium were adjusted prior to autoclaving the medium and were measured again at the end of the experiment to ensure that the pH had not changed during the experiment. The nutrient tests were made in duplicates depending upon the nutrient being tested. The effect of different nutrient sources on the growth was tested for both the presence and the absence of the corresponding substrate. The growth of the cultures was monitored by turbidity measurements at 600 nm using the DEN-600 photometer (Biosan SIA, Latvia). For some experiments, the closely related phototrophic bacterium *G. groenlandica* TET16 (DSM 110,279 ^T^) was grown for comparison.

### 2.2. Transcriptome Sequencing and Analysis

Cells were harvested by centrifugation (8000 × *g* for 3 min, 4 °C). Pellets were resuspended in 1 mL PGTX extraction solution [18] and immediately frozen in liquid nitrogen. Cells were transferred into 2 mL Eppendorf tubes and immediately frozen in liquid nitrogen and stored at –80 °C until extraction. RNA was extracted as described earlier [19]. The RNeasy kit (Qiagen, the Netherlands) was used for purification according to the manufacturer’s protocol. The first digestion of genomic DNA was performed on the column, using DNase I (Qiagen, The Netherlands) as described by the manufacturer. Total RNA was eluted in 88 μL RNase-free H_2_O, and the second DNase I digestion was made in solution, followed by a second RNeasy purification step, which included an additional washing step with 80% ethanol done before elution with 30 μL RNase-free water.

Libraries were generated according to Shishkin et al. [20] including rRNA removal with the RiboZero Kit (Illumina Inc., San Diego, CA, USA). The library was sequenced on a NovaSeq 6000 (Illumina Inc., USA) in paired-end mode with 100 cycles in total using the FASTQ-mcf suite (https://github.com/ExpressionAnalysis/ea-utils accessed on 20 December 2021). The image analysis and base calling were performed using the Illumina pipeline v 1.8 (Illumina, San Diego, CA, USA). Raw reads were processed, and differential gene expression was assessed as described before [19]. Low-quality bases (Phred score < 30) and Illumina adapters were clipped. Briefly, quality filtered reads were mapped to the AP64 genome (NCBI RefSeq accession GCF_000695095.2) using bowtie2 [21]. FeatureCounts was used to assess the number of reads *per* gene [22]. Normalization and identification of significantly differentially regulated genes (FDR < 0.01, absolute log2 fold change (log2FC) > 1) were performed with edgeR [23]. The heatmap was generated with the package pheatmap. Hierarchical clustering based on the Euclidian distance of log2FC data was used to cluster genes. The obtained tree was cut into three clusters based on the branching points.

### 2.3. Procedure for Acid Digestion of Cells and Elemental Analysis with Inductively Coupled Plasma Mass Spectrometry (ICP-MS)

The 40 mL of cell culture was collected onto mixed cellulose ester (MCE) filters (Millipore, porosity 0.45 µm, diameter 25 mm) using gentle vacuum. The filter was washed by an equal amount of distilled water and placed in a desiccator containing silica gel. Once it was completely dry, the weight of the sample was determined by subtracting the initial weight of the MCE filter from the final weight of the MCE filter. Due to handling of small amounts of biomass, it was decided to digest the MCE filter membrane. MCE filter membranes were chosen as these can be digested easily with the protocol mentioned in Andresen et al. [24] contrary to other types of filter membranes. The filters were weighed with and without biomass and digested with a mixture of 85 mL/100 mL of 70% HClO_4_ (Suprapur^®^ grade, Carl Roth, Karlsruhe, Germany) and 15 mL/100 mL of 69% HNO_3_ (Ultrapur^®^ grade, Karlsruhe, Germany) in Duran glass tubes as mentioned in Andresen et al. [24]. The digestion was done using a Fuji PXG4 Thermoblock (AHF Analysentechnik AG, Germany), and after evaporation of the acid mix and cooling down, 0.5 mL of 5% HCl (Ultrapur^®^ grade, Carl Roth, Karlsruhe, Germany) was added to each test tube to re-dissolve the salts. The glass tubes were heated to 90 °C for 1 h to obtain clear solutions. The final volume of 1.5 mL was set with ddH_2_O. Appropriate dilutions were done with 0.2% HNO_3_. Indium was added as an internal standard at 1 ng mL^−1^ to each test solution. The ICP multi-element standard solution VI (Merck KGaA Darmstadt Germany) was used to prepare standard curves. Analyses were done using the sector field ICP-MS (ICP-sfMS) Element XR-2 with a jet interface (Thermo Fisher Scientific, Bremen, Germany). Triplicate measurements of each technical replicate were made with lowest possible relative standard deviations. The instrument was tuned for least oxide ratios and maximum sensitivity. The potential interferences on the analyte of interest Ca^44^ were ^27^Al^17^O, ^26^Mg^18^O, ^12^C^16^O^16^O, ^14^N^14^N^16^O, ^28^Si^16^O etc., mainly eliminated with selection of high resolution 10000, which successfully resolves all these possible interferences. Other interferences on other analytes were dealt with medium resolution 4000.

### 2.4. Transmission Electron Microscopy

For transmission electron microscopy (TEM), the sample was concentrated by centrifugation at 3000 rpm for 3 min. Sample (4 μL drop) was applied to carbon coated copper grids (mesh 400) for 1 min followed by washed in two drops of water. Then a 4 μL drop of 1.2% aqueous uranyl acetate solution was applied on the grid for another 1 min and dried. The grids were visualized using Jeol JEM 1400 microscope at an operating voltage of 120 kV for detecting the cell morphology and flagella.

### 2.5. Statistical Analysis

Data analyses were performed using Sigma Plot (version 14.0). The data were analyzed for normal distribution using the Kolmogorov–Smirnov test. Multiple comparisons were carried out by one-way analysis of variance (ANOVA) and Tukey’s *post hoc* test (parametric data), Kruskal–Wallis and Dunn’s *post hoc* tests were used for the non-parametric data. The results are presented as mean ± SD, and *p*-values < 0.05 were considered statistically significant.

## 3. Results

### 3.1. Medium Optimization

There may be two possible explanations as to why *G. phototrophica* did not grow in liquid medium: (1) *G. phototrophica* is an organism naturally growing in biofilm and thus prefers the growth on solid medium, or (2) agar provides some vital nutrients, which are not present in the current formulation of the liquid medium. To test these two hypotheses, we grew the cells on a solid nutrient medium containing either regular bacteriological agar or pure agarose for molecular biology. Pure agarose was used since it does not contain any undefined nutrients, but it still serves as a gelling agent providing a support for the bacterial biofilm.

The agar and agarose plates were inoculated and incubated for 14 days in a desiccator in the dark under reduced oxygen tension (10% O_2_). *G. phototrophica* grew on the agar but not on the agarose plates. This result indicated that the agar likely provides some vital nutrient(s) for *G. phototrophica* growth. Therefore, we performed a systematic search for the missing nutrient(s) in the growth of *G. phototrophica* on the agar plates. First, we tested growth on different agars. The growth of the *G. phototrophica* cells was found to be best on Bacto^TM^ Agar followed by Micro agar and Plant agar. Utilization of different compounds as a carbon source was assayed on the agar plates containing Bacto^TM^ Agar. Carbohydrates like rhamnose, trehalose, erythritol, adonitol, melibiose, and dulcitol were tested. We noticed that the growth of the *G. phototrophica* cells was either not improved or delayed. Vitamins like folic acid, para-aminobenzoic acid, niacin, thiamine, riboflavin, nicotinamide, cobalamin, and pyridoxine hydrochloride were tested on Bacto^TM^ Agar plates. It was found that these vitamins were not indispensable sources but played a role in improving the growth of the *G. phototrophica*. With the results obtained from culturing on the agar plates with different combination of nutrient sources, we started to culture the cells in the modified liquid medium containing the nutrients which had the best effect on the growth. Since medium for cultivation of phototrophic bacteria often contains additional elements which are not part of the original R2A^+^ medium, we tested them separately. The inorganic components such as Co^2+^, Sr^2+^, SiO_3_^2-^, Mn^2+^, Ni^2+^, Zn^2+^, Cu^2+^, and Ca^2+^ were added to test the growth. The strongest effect was observed with calcium, where the addition of 50 mg L^−1^ CaCl_2_ into the liquid medium resulted in an ample growth of *G. phototrophica* cells within 7 days (Figure 1). We also determined the optimal growth pH and temperature for growth in liquid medium in a different temperature and pH range. We found that the optimal growth temperature is between 26–28 °C and the optimal pH is 7.3–7.5.

### 3.2. Calcium Requirement

We tested the growth of *G. phototrophica* in liquid medium with different concentrations of CaCl_2_ to detect the minimum and optimum concentration of CaCl_2_. Addition of 2 mg L^−1^ CaCl_2_ was sufficient to induce the stable growth of the cells. The best growth of the cells was observed at 20 mg and 50 mg L^−1^ CaCl_2_ (Figure 2A). Higher CaCl_2_ concentrations were not tested, since they caused the precipitation of the phosphate present in the medium. We also conducted the same control experiment with *G. groenlandica*, a close relative of *G. phototrophica*. However, this bacterium grew well even in liquid medium without added calcium (Figure 2B), which showed that *G. phototrophica* requires higher calcium concentrations for its growth when compared to other members of *Gemmatimonas* genus.

The morphology of *G. phototrophica* cells was observed by electron microscopy. The cells grown in 20 mg L^−1^ CaCl_2_ had dimensions of 2.25 × 0.7 µm and had a single flagellum with a length ranging between 3.9 to 7.5 µm (Figure 3). In contrast, the cells grown on low calcium medium were elongated with dimensions of 3.9 × 0.6 µm, and the flagella were frequently missing (Figure 3). The *G. groenlandica* cells grown in 20 mg L^−1^ CaCl_2_ and 50 mg L^−1^ CaCl_2_ were rod-like, whereas the *G. groenlandica* cells grown in 2 mg L^−1^ CaCl_2_ were elongated. Interestingly, its flagella were fully developed and up to 10 µm long in all the cases (Figure 3).

To establish the specificity of the calcium effect, we performed an experiment where calcium was substituted with either magnesium or strontium, which are also divalent cations from the same group in the periodic table of elements. While addition of magnesium did not have any strong effect (magnesium was a part of the original medium composition), strontium was able to support growth of *G. phototrophica* in liquid medium. We observed that the addition of SrCl_2_ facilitated the growth of *G. phototrophica*, although the growth rate was significantly lower when compared to medium amended with calcium (Figure 4).

Calcium has two main roles in various organisms. Either it is required to build inorganic structures in the cells such as shells or skeletons, or it is required for more specific purposes like enzymatic actions or assembly of specific biological complexes. To discriminate between these two options, we measured the amount of calcium in *G. phototrophica* biomass using ICP-MS. The content of calcium in the cells increased with the amount of calcium in the culture medium ranging from 6.76 × 10^−3^ to 8.79 × 10^−3^ calcium per DW (*w:w*). Interestingly, the calcium content in control cells of *G. groenlandica* was more similar to the *G. phototrophica* cells cultured in low calcium (Figure 5A). This result suggests that *G. phototrophica* requires somewhat higher cellular levels of calcium for its physiology and metabolism. The ICP-MS analysis furthermore documented that the relative content of strontium was higher in the cells grown in the calcium-limited medium (Figure 5B), which is consistent with the observation that strontium can partially substitute for calcium.

### 3.3. Transcriptome Response to Calcium Amendment

To identify which genes are activated after calcium amendment, we grew *G. phototrophica* in medium containing 2 mg L^−1^ CaCl_2_. After 3 days, the cultures were amended with 20 mg L^−1^ CaCl_2_, or 20 mg L^−1^ MgCl_2_ (Figure 6A). The gene expression profiles were analyzed for all treatments before as well as 2, 8, and 24 h after the respective amendments (Appendix A).

The obtained sequencing depth varied between 350,000 and 15 Mio. with a median of 7.5 Mio. reads. For data quality control and to check the reproducibility of independent replicates, we performed multidimensional scaling (MDS). The samples grown with different concentrations of CaCl_2_ and MgCl_2_ followed distinct trajectories separating them by time on the *x*-axis and by amendment on the *y*-axis (Figure 6B). We identified in total 203 differentially expressed genes (DEGs), in which 74 were upregulated and 129 were downregulated when comparing 20 mg L^−1^ CaCl_2_ vs. 2 mg L^−1^ CaCl_2_. A total of 67 DEGs were identified when comparing 20 mg L^−1^ MgCl_2_ vs. 2 mg L^−1^ CaCl_2_, in which 4 were upregulated and 63 were downregulated. A complete list of DEGs can be found in Appendix A.

Genes associated with membrane transport were differentially regulated in 20 mg L^−1^ CaCl_2_ at 2 h, 8 h and 24 h. Cells grown in 20 mg L^−1^ CaCl_2_ at 2 h and 24 h showed a similar number of DEGs and a relatively similar log2 fold change (log2FC). There was an overlap of 47 DEGs between 2 h and 24 h, whereas 8 h showed a little overlap with 2 h and 24 h in the cells grown with 20 mg L^−1^ CaCl_2_ (Figure 6C). There was no overlap of the DEGs between time points in the cells grown with 20 mg L^−1^ MgCl_2_. However, there was an overlap of 4 DEGs between the cells grown with 20 mg L^−1^ MgCl_2_ and 20 mg L^−1^ CaCl_2_ at 2 h (Figure 6C).

One locus (GEMMAAP_RS18425-18450) containing the respiratory enzymes involved in oxidative phosphorylation such as the cytochrome *c* oxidase accessory protein (CcoG), cbb3−type cytochrome oxidase assembly protein (CcoS), *c*−type cytochrome and cbb3−type cytochrome *c* oxidase subunit 3, was significantly upregulated in the 20 mg L^−1^ CaCl_2_ compared to 20 mg L^−1^ MgCl_2_ at 8 h. Also, another locus (GEMMAAP_RS19165-19180) containing genes involved in tetrapyrrole synthesis such as the oxygen independent coproporphyrinogen III oxidase, *bchJ*, magnesium protoporphyrin IX monomethyl ester anaerobic oxidative cyclase, *bchE*, and protoporphyrinogen oxidase, *hemJ* outside of the PGC, was significantly upregulated in the 20 mg L^−1^ CaCl_2_ compared to 20 mg L^−1^ MgCl_2_ at 8 h. Interestingly, all these genes were downregulated at 2 and 24 h, respectively (Figure 7). In contrast, the genes encoding enzymes for bacteriochlorophyll synthesis and encoded in the PGC showed a slightly decreased expression with CaCl_2_ only.

In particular, we were interested in the possibility of knowing the key role of calcium ions in the regulation of the calcium transport pathway of *G. phototrophica*. The chemotaxis protein CheA (GEMMAAP_RS15835) and methyl-accepting chemotaxis protein (GEMMAAP_RS00390), the integral membrane protein involved in the flagellar-motor complex showed a specific upregulation in 20 mg L^−1^ CaCl_2,_ whereas both genes were downregulated in 20 mg L^−1^ MgCl_2_ at 8 h after amendment. The response to the 20 mg L^−1^ CaCl_2_ amendment was mostly acute and restricted to 8 h (Figure 7). The DEGs were associated with the organization of cellular components and membrane transport. We also found that the heme proteins and the two genes coding for the cation-transporting P-type ATPase involved in proton pumping (GEMMAAP_RS13065 and 18445) were significantly upregulated at 8 h for 20 mg L^−1^ CaCl_2_ compared to 20 mg L^−1^ MgCl_2_ (Figure 7).

## 4. Discussion

The objective of the presented study was to optimize the medium composition in order to establish liquid cultures of *G. phototrophica*. During the process, we found that its growth was primarily restricted by the lack of calcium. Besides, we did not find any other nutrient source bolstering its growth in liquid as explained in the results section. The established threshold of 20 mg L^−1^ CaCl_2_ corresponds to 7.2 mg Ca^2+^ L^−1^. In a recent report, it is described that the global median concentration of calcium in freshwaters is 4 mg L^−1^ [25]. The study documented that calcium concentrations are strongly linked to alkalinity, with the highest calcium and carbonate levels found in freshwaters with a pH around 8.0. In contrast, acidic lakes also have low Ca^2+^ concentrations, which may seriously limit the growth of *G. phototrophica* in these waters.

The higher calcium requirement also explains why *G. phototrophica* could have been grown on agar media [8,26]. Agar itself contains approximately 1 g of calcium per kg, which for 2% agar results in 20 mg calcium per L of agar medium. This concentration was fully sufficient to support growth. In contrast, the calcium content in peptone was almost nil and in yeast extract was only ~0.5 g per kg, which did not provide a sufficient enough amount to support the growth in liquid medium.

Intense research documented that calcium has multiple roles in eukaryotic organisms. Calcium plays a central role in various eukaryotic cellular functions and processes, predominantly via the action of cell signaling. Such roles include cell motility, proliferation, growth, calcification, flagella development, neurotransmission and many other biological processes [27,28,29,30]. Cells have to adapt to environmental changes by signal transduction and calcium ions often play the role of a versatile messenger, which transmits signals from the cell surface to the interior of the cells [27]. Calcium signaling is forced through the presence of concentration gradients across cellular membranes, as there are fewer calcium ions within the cell compared to extracellular concentrations [31]. However, excess calcium within the cytosol can cause subsequent deleterious effects, therefore a low cytosolic calcium concentration is ideally maintained by the action of calcium pumps, the process of chelation, or calcium compartmentalization in eukaryotes [32].

Much less is known about the specific role of calcium in prokaryotic cells. Calcium is required for the bacterial cytochrome *c* oxidase [33,34,35], which is in agreement with our finding that genes for this enzyme were up-regulated when more calcium was supplied. It is reported that calcium ions may play an essential role in chemotaxis, cell cycle, and competence by using calcium ions from an external source and calcium antagonists in prokaryotes [36,37,38]. However, direct evidence of the role of intercellular calcium ions in prokaryotes is still poorly understood and remains to be elucidated [39,40]. It is also reported that calcium impacts the control of gene expression and cytoskeletal reorganization in bacterial pathogens [41]. Bacteria maintain cytosolic calcium. It has previously been shown that the free cytosolic calcium levels maintained by the prokaryotic cells are quite low (100–300 nM) [42,43]. However, the molecular mechanisms by which the calcium levels are maintained remains unanswered

In the presented study we have observed that cells grown in low calcium concentrations were frequently elongated, which signaled possible problems with cell division and formation of the cytoskeletal complexes. We found many genes that are involved in cell division to be upregulated in the 20 mg L^−1^ CaCl_2_ compared to the 2 mg L^−1^ CaCl_2_. There are reports on the formation of elongated or filamentous cells in many species such as *B. subtilis* instead of the bloated and spherical cells in the absence of cytoskeletal proteins involved in the assembly of the cytoskeletal complexes [44]. It is apparent that cells grown in low calcium concentration lack flagella compared to cells grown in high calcium concentration as mentioned in the results. We found the chemotaxis and *fli* genes which are involved in flagella formation were slightly upregulated in cells grown in high calcium concentration.

We found that the calcium content was about 20% higher in the cells grown in 20 mg L^−1^ CaCl_2_ when compared to the cells grown in 2 mg L^−1^ CaCl_2_. This suggests that the *G. phototrophica* cells require higher calcium concentration in their cells. Recent reports showed that calcium has control over the formation and activity of the cytoskeleton by modulating the proteins associated with the structural organization [30]. The growth and division of bacterial cells are tightly coordinated with the cytoskeleton to maintain its structural integrity [45,46]. The cytoskeleton gives an order to a cell, and in large cells, where diffusion may become limiting, it facilitates the transport of the metabolites in the cell [47]. Some bacteria have the capability to store calcium ions in membrane-bound structures. Since it has been reported in *E. coli* that the cytosolic free calcium is regulated through influx and efflux [39], we aimed to identify putative genes involved in calcium efflux pathways. It was expected that proteins involved in calcium efflux will be upregulated in the cells grown at higher concentrations of calcium, whereas those involved in calcium influx would be downregulated. However, we did not find the regulation of proteins involved in calcium signaling in this study.

The performed transcriptome analysis documents upregulation of genes linked to cell division, RNA synthesis, photosynthesis and respiration in the 20 mg CaCl_2_ L^−1^ compared to 20 mg MgCl_2_ L^−1^. There are several studies indicating that calcium ions play a regulatory role in the physiology of prokaryotes [39,48,49,50,51,52]. The transcriptomic analyses in *Escherichia coli* and *Bacillus subtilis* revealed that the expression of hundreds of genes is regulated by changes in calcium ions in response to increased calcium levels from external sources [39,50]. They reported that the processes such as flagella formation, biofilm matrix production, iron acquisition, polysaccharide production and general stress response were affected. We also found that genes such as calcium transporting ATPase and large conductance mechanosensitive channel MscL, to which calcium is specifically bound and acts as cofactor, were not significantly upregulated in high calcium.

## 5. Conclusions

In conclusion, the presented study shows that *G. phototrophica* requires a higher concentration of calcium than related microorganisms for its growth in liquid medium. While calcium is known to be required, e.g., for bacterial respiration due to its role in cytochrome oxidase, the elevated requirement here was unexpected and furthermore affected the regulation of genes involved in tetrapyrrole biosynthesis, cell division, transition metal transport, and other fundamental cellular pathways. Yet, the exact role of calcium in *G. phototrophica* still has to be determined. Now with the established liquid growth conditions, much more detailed research on its physiology, regulation, and metabolism can be undertaken.

## Figures and Tables

**Figure 1 microorganisms-11-00027-f001:**
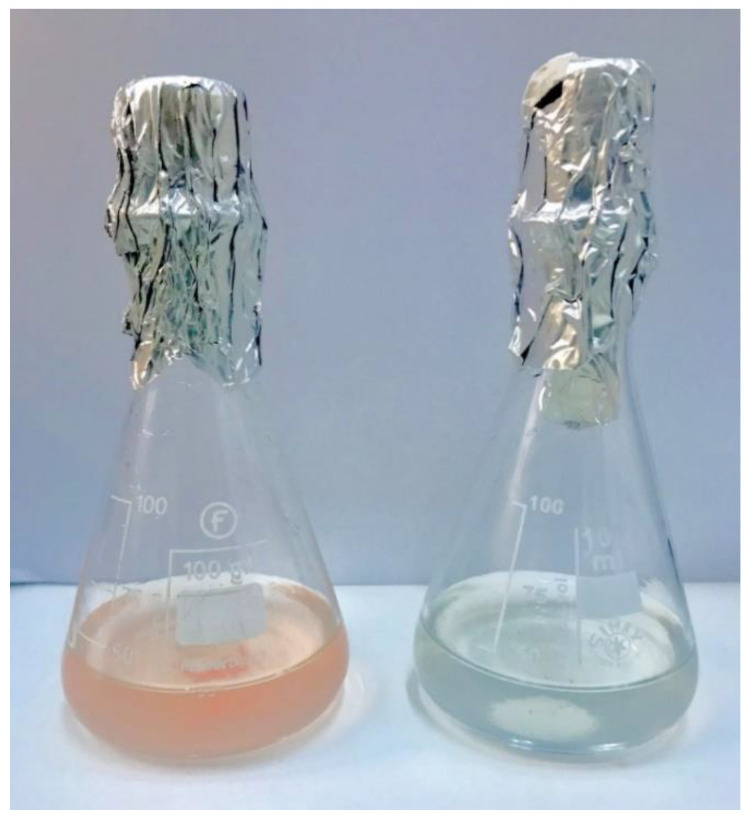
Liquid culture of *G. phototrophica* showing growth in the medium containing 50 mg CaCl_2_ L^−1^ (**left** flask) and no growth in medium without added CaCl_2_ (**right** flask).

**Figure 2 microorganisms-11-00027-f002:**
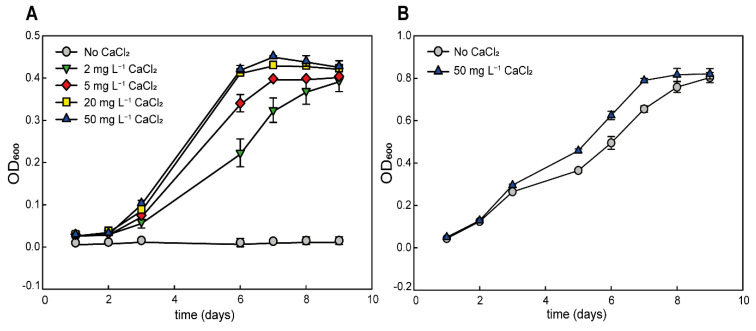
(**A**) Growth rate of *G. phototrophica* grown in liquid media with different CaCl_2_ concentrations. (**B**) Growth rate of *G. groenlandica* in medium with 50 mg CaCl_2_ L^−1^ and in medium without added CaCl_2_.

**Figure 3 microorganisms-11-00027-f003:**
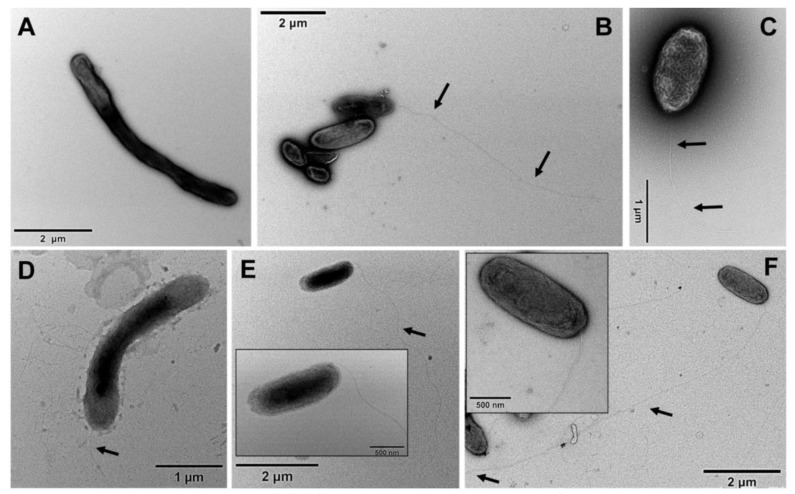
Transmission electron microscopy images of *G. phototrophica* and *G. groenlandica* cells grown in with and without CaCl_2_. (**A**) Cells of *G. phototrophica* in 2 mg CaCl_2_ L^−1^ were long rod shaped without flagella. (**B**) Cells of *G. phototrophica* in 20 mg CaCl_2_ L^−1^ were short rod shaped with polar flagella. (**C**) Cells of *G. phototrophica* in 50 mg CaCl_2_ L^−1^ were rod shaped with flagella. (**D**) Cells of *G. groenlandica* in 2 mg CaCl_2_ L^−1^ were rod-like and elongated in shape with short flagella. (**E**) Cells of *G. groenlandica* in 20 mg CaCl_2_ L^−1^ were rod shaped with flagella. (**F**) Cells of *G. groenlandica* in 50 mg CaCl_2_ L^−1^ were rod shaped with long flagella. Scale bar is included in each figure. Inlet shows the close-up images. Arrows show the flagella.

**Figure 4 microorganisms-11-00027-f004:**
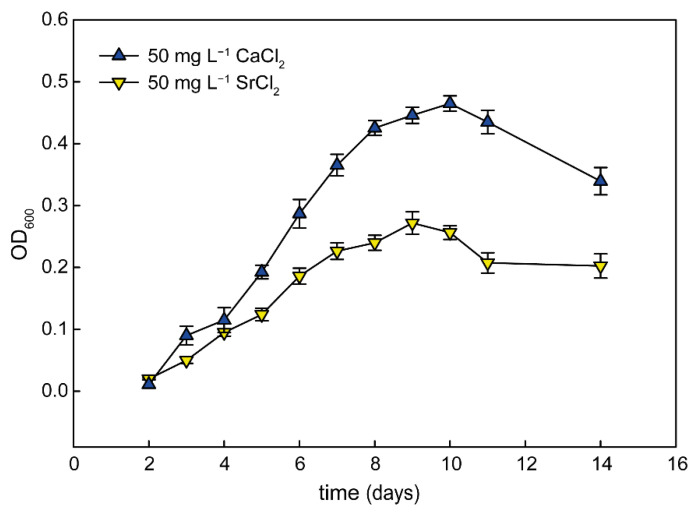
Growth rate of *G. phototrophica* grown in liquid media with 50 mg L^−1^ CaCl_2_ and 50 mg L^−1^ SrCl_2_.

**Figure 5 microorganisms-11-00027-f005:**
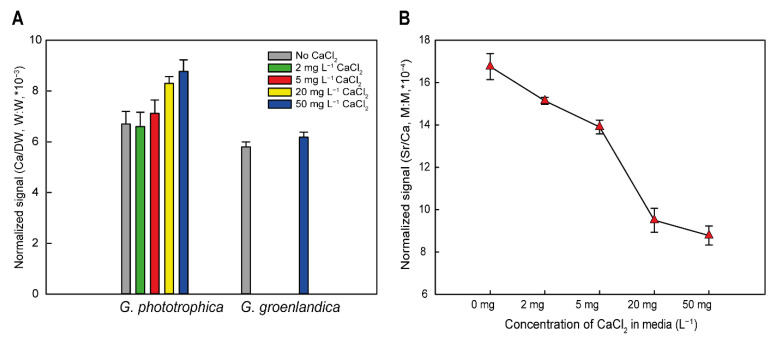
(**A**) Accumulation of calcium in *G. phototrophica* and *G. groenlandica* cells grown in nutrient media containing different concentrations of CaCl_2_. (**B**) Accumulation of strontium in *G. phototrophica* cells grown in nutrient media containing different concentrations of CaCl_2_.

**Figure 6 microorganisms-11-00027-f006:**
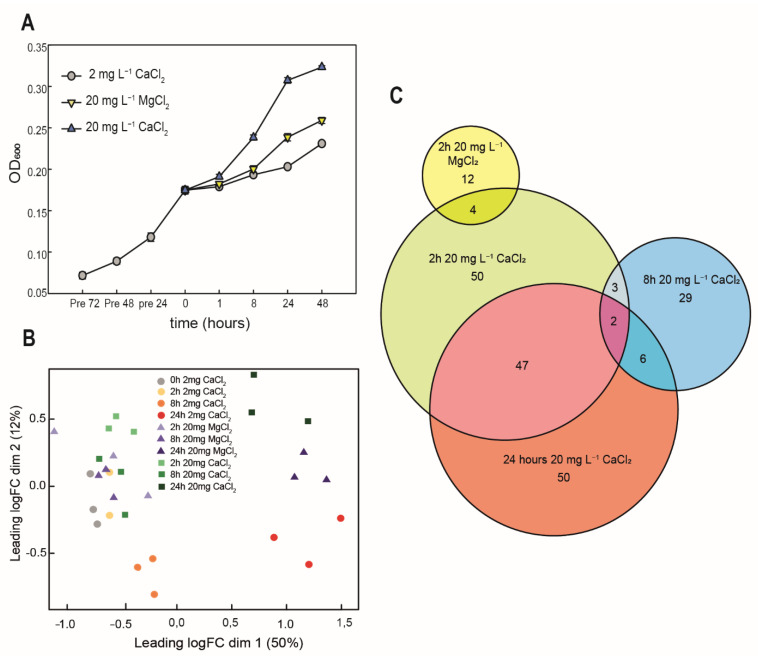
(**A**) The figure shows the growth rate of *G. phototrophica* grown in 20 mg CaCl_2_ L^−1^ and no growth in 2 mg CaCl_2_ L^−1^ (Note: scale bar in the *x*-axis is uneven). (**B**) Multi-Dimensional Scaling (MDS) plot showing the similarity between the treatment replicates. (**C**) Venn diagram showing the differentially expressed genes in 20 mg MgCl_2_ L^−1^ and 20 mg CaCl_2_ L^−1^ at 2 h, 8 h and 24 h after amendment.

**Figure 7 microorganisms-11-00027-f007:**
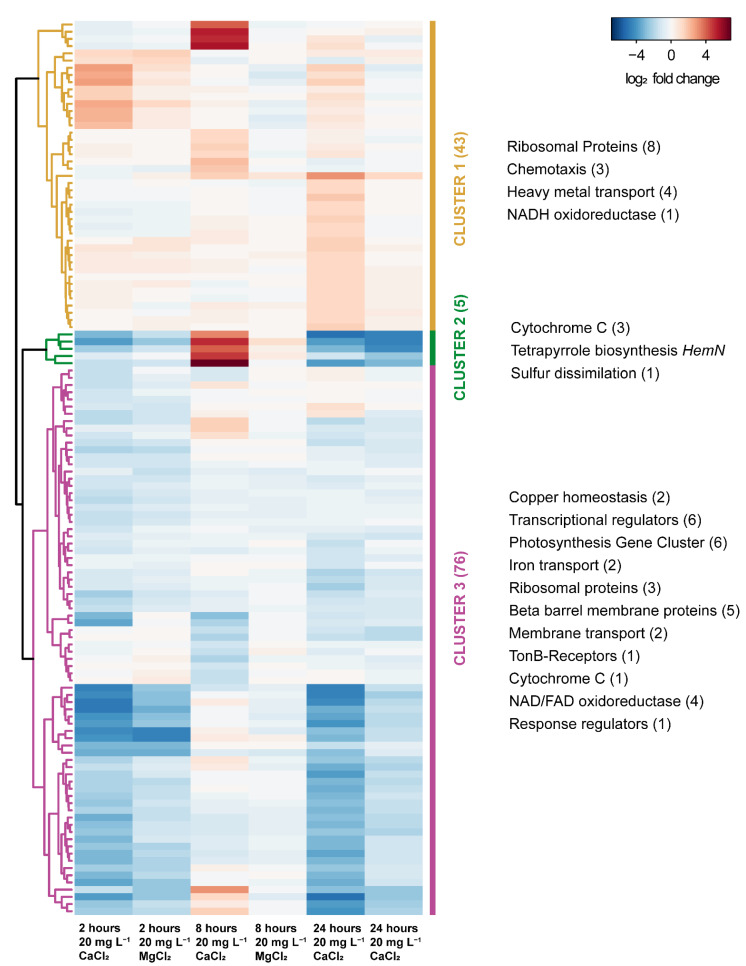
Heatmap and hierarchical clustering based on the Euclidean distance of the log2 FC of significantly differentially expressed genes in 20 mg CaCl_2_ L^−1^ and 20 mg MgCl_2_ L^−1^ at 2 h, 8 h, and 24 h after amendment. The colors indicate three clusters obtained by cutting the tree at the earliest branching points. Functions of genes within the clusters are noted on the left with the numbers of coding genes in brackets.

## Data Availability

Data availability. RNA sequencing data is publicly available at the NCBI gene expression omnibus database under accession number GSE217809 (https://www.ncbi.nlm.nih.gov/geo/query/acc.cgi?acc=GSE217809, accessed on 20 December 2022).

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
