# Peer review of "The Influence of Calcium on the Growth, Morphology and Gene Regulation in Gemmatimonas phototrophica"

_microorganisms, 2022, doi:10.3390/microorganisms11010027_

Round 1
Reviewer 1 Report
The manuscript by Shivaramu et al., entitled “The influence of calcium on the growth, morphology and gene regulation in Gemmatimonas phototrophica” (microrganisms-2087315), describes a detailed study on the liquid cultivation of G. phototrophica. The authors disclosed the reason why G. phototrophica grew only on agar plates in the previous studies. This study is valuable because Gemmatimonas represents the sole phototrophic genus in the phylum Gemmatimonadetes and the large-scale cultivation of the bacterium is crucial for further biochemical characterization. The combination of cultivation, transcriptome, and microscopic experiments captures the effect of calcium amendment on G. phototrophica cells. As there exists relatively little earlier literature about the calcium requirement for bacterial growth, this manuscript will represent a significant contribution to the field.
Yet, prior to publication, the authors need to further address the points mentioned below.
In section 2.1, the authors should explain more on the cultivation conditions like they describe in the first section of Results.
L251 “3.2. Transcriptome response to calcium amendment” should be 3.3. Transcriptome~.
In this section, locus tags of genes and EC numbers should be added for the proteins that were mentioned in the context.
L34 The genus name should be spelled out because this is the first appearance in the main text.
L237 ICP and ICP-MS are not the first appearance here, they are at L129-130.
L256 Summary information of transcriptomic analysis should be described here. How many reads were obtained? How many genes out of all CDSs were significantly expressed in each culture condition? etc...
L290-292 Here, the authors mentioned that no gene for bacteriochlorophyll biosynthesis in PGC responded to the amendment, but, in Figure 7, one gene in PGC seemed to respond. What is the gene for?
The caption for Figure 7 does not seem to be enough. Please add more explanations.
Reference 26 Is this a correct citation?
In figures showing bacterial growth, logarithmic y-axis should be used.
Reviewer 2 Report
The authors provided a detailed and interesting analysis of the effect of calcium on the growth and gene regulation of Gemmatimonas phototrophica. G. phototrophica is a fairly newly discovered organism and was only able to be cultivated on agar plates and had failed to grow in liquid cultures until the author’s discovery. This had hampered further study and analysis of this unique Gemmatimonadota species.
The paper is well constructed and provides convincing results and discussions about the role of calcium on the G. phototrophica growth. It would however benefit from a few additions or changes before publication:
- Section 2.2, lines 102-109. More detail should be provided about the bioinformatics used and the parameters used to generate the specific gene clusters. The authors do refer to another transcriptomics paper (ref 19), however the current paper would benefit from some more extensive description here on read processing and manipulation, data curation and possible cutoff parameters, etc.
- Line 26: higher concentration than what, or compared to what? Compared to other Gemmatimonas species? Please explain here.
- Lines 221-228: (figure 3 legend): ‘were rod in shape’ sounds awkward, perhaps use ‘rod shaped’ instead.
- Line 228: change ‘shows’ to ‘show’ (there are multiple arrows per figure)
- Line 322: delete ‘)’
- In general, the authors did not explain how they came to the hypothesis to focus on calcium as the determining factor. What else was tested from the original media and was calcium the only factor? A short expansion on this in the background or discussion section would be helpful.
